# A new lidar inversion method using a surface reference target. Application to the backcattering coefficient and lidar ratio retrievals of a fog-oil plume at short-range

Florian Gaudfrin[1,2,3], Olivier Pujol[2], Romain Ceolato[1], Guillaume Huss[3], and Nicolas Riviere[1]

[1]ONERA / DOTA, Université de Toulouse, F-31055 Toulouse - France
[2]Université de Lille, Département de physique, Laboratoire d'optique atmosphérique, 59655 Villeneuve d'Ascq, France
[3]LEUKOS, 37 rue Henri Giffard, 87280 Limoges, France

**Correspondence:** Florian Gaudfrin (florian.gaudfrin@onera.fr)

**Abstract.** In this paper, a new elastic lidar inversion equation is presented. It is based on the backscattering signal from a surface reference target (SRT) rather than that from a volumetric layer of reference (Rayleigh molecular scatterer) as usually done. The method presented can be used when the optical properties of such a layer are not available, *e.g.* in the case of airborne elastic lidar measurements or when the lidar-target line is horizontal Also, a new algorithm is described to retrieve the lidar ratio and the backscattering coefficient of an aerosol plume without any *a priori* assumptions about the plume. In addition, our algorithm allows a determination of the instrumental constant. This algorithm is theoretically tested, *viz.* by means of simulated lidar profiles, and then using real measurements. Good agreement with available data in the literature has been found.

## 1 Introduction

Atmospheric aerosols are liquid or solid particles dispersed in the air (Glickman and Zenk, 2000) of natural (volcano, biomass burnings, desert, ocean) or anthropogenic origins. They play an important role in cloud formation (DeMott et al., 2010), radiative forcing (Hansen et al., 1997)(Hay, 2000) and more generally for researches on the climate change, but also in the context of air quality and public health (Bal, 2008; Finlayson-Pitts and Pitts, 2000; Popovicheva et al., 2019; Zhang et al., 2018). Their size varies from the nanometer to the millimeter scale (Robert, n.d.). However, a large majority of aerosols have a size between $0.01\,\mu m$ and $3\,\mu m$ (Clark and Whitby, 1967) for which scattering is dominant in the optical domain. The Mie theory is often used, at least statistically (*i.e.* for a large population of random sized aerosols), although aerosols are not always spherical. The optical backscattering and extinction properties of aerosols are mainly related to their shape (Ceolato et al., 2018), size distribution (Vargas-Ubera et al., 2007), concentration and chemical composition which is based to their nature (dust, maritime, urban). Lidar are active remote sensing instruments suitable for aerosol detection and characterization (Sicard et al., 2002) over kilometric distances during both day and nighttime.

The optical properties of aerosols are obtained by means of inversion methods using the simple scattering lidar equation. In the 1980s, a stable one-component formulation adapted to lidar applications was proposed by Klett (1981). It has then been extended to a two-component formulation, *viz.* separating molecular and aerosol contributions, by Fernald (1984) and Klett (1985). The elastic lidar equation is an ill-posed problem in the sense of Hadamard (1908) since one searches for extinction and for backscattering coefficients with only a single observable. Several assumptions are therefore required in order to invert the lidar equation:

(i) A calibration constant is usually determined from a volumetric layer of the upper atmosphere as a reference target (Vande Hey, 2014). This calibration layer can be very high in altitude; it has recently been moved from the around $(32\,\mathrm{km})$ to around $(36 - 39\,\mathrm{km})$ for the CALIPSO spaceborne lidar in order to reduce uncertainties in the inversion procedure (Kar et al., 2018; Getzewich et al., 2018). This volume is considered made only of pure molecular constituents whose optical scattering properties are well-known (Rayleigh regime). The molecular backscattering coefficient is generally estimated from the standard model of the atmosphere (Anon, 1976; Bodhaine et al., 1999). However, poor estimates of the reference or low signal-to-noise ratios $(SNR)$ can lead to severe uncertainties on the retrieved extinction and backscattering coefficients. Few sensitivity studies have been performed to evaluate such uncertainties (Matsumoto and Takeuchi, 1994; Rocadenbosch et al., 2012). Spatial averaging around the volume of reference in addition to time averaging is thus recommended to increase $SNR$.

(ii) Lidar ratio is constant over the distance range of measurements (Sasano et al., 1985). This is also a source of important errors in the retrieval values. Some studies have proposed a variable lidar ratio under the form of a power-law relationship between the extinction and backscattering coefficients, but such a method requires an *a priori* knowledge of the medium under study (Klett, 1985).

(iii) The molecular contribution along the lidar-line is known. It is estimated, as for the backscattering coefficient, by means of temperature and pressure vertical profiles, using either the standard model of the atmosphere or radio soundings (Jäger, 2005).

In the case of elastic lidar inversion, the most critical parameter is the lidar ratio $(LR)$. It depends on the wavelength (in vacuum) and on the microphysics, morphology, and size of the particles (Hoff et al., 2008). The $LR$ ranges from $20\,\mathrm{sr}$ to $100\,\mathrm{sr}$ at $532\,\mathrm{nm}$ (Ackermann, 1998; Cat, 2005; Leblanc et al., 2005) according to the aerosol origins (maritime, urban, dust particles, biomass burning). It is therefore difficult to assume an *a priori* value for $LR$ in as much this information is to be found rather than given.

Several alternatives have been analyzed to constrain the inversion procedure while relaxing assumption (ii). These alternatives are based on the determination of the optical thickness, the one which consists in coupling lidar and sunphotometer measurements being the most largely used. The measured optical thickness is then used to constrain extinction profiles (Fernald et al., 1972; Pedrós et al., 2010). A second alternative, consists in combining elastic lidar and Raman measurements in order to get the optical depth as a function of range (Ansmann et al., 1990, 1992, 1997; Mattis et al., 2004). In a third technique, the

optical depth is retrieved from elastic lidar measurements with different zenith angles (Sicard et al., 2002). It is worth indicating that coupling lidar and sunphotometer measurements is possible only daytime while Raman measurements are carried out preferentially at nighttime in order to increase the $SNR$. A fourth method consists in the determination of the optical thickness and lidar ratio of transparent layers located above opaque clouds (Hu et al., 2007; Young, 1995) that are used as reference for calibration in the inversion procedure (O'Connor et al., 2004). This method is used for downlooking lidar measurements capable of measuring depolarization ratios. However, the method is limited to lidar systems in non-polarized detection and for lidar measurements for which clouds cannot be used as a reference. A fifth approach consists in the determination of the optical thickness of the atmosphere from the sea surface echo by combining lidar and radar measurements (Josset et al., 2010a, b, 2008). This method has been used to find the lidar ratio and the optical depth of aerosol layers over oceans (Dawson et al., 2015; Josset et al., 2012; Painemal et al., 2019).

Another limitation of ground-based lidar measurements is related to the overlap function that strongly impacts (and prevents) observation close to the instrument, *i.e.* in the lowest layers of the troposphere where aerosols are emitted. Different studies have proposed to modify the overlap function analytically (Comeron et al., 2011; Halldórsson and Langerholc, 1978; Kumar and Rocadenbosch, 2013; Stelmaszczyk et al., 2005) or empirically (Vande Hey et al., 2011; Wandinger and Ansmann, 2002). Some lidar devices are also equipped with a second telescope of higher overlap at short range (Ansmann et al., 2001). However, current lidar systems are not adapted enough to the monitoring and characterization of volumetric targets at short-range, for instance in the industrial context, or more generally, for anthropogenic activities (Ceolato and Gaudfrin, 2018).

To meet new industrial emission control requirements and very recently emitted anthropogenic aerosols characterization, we have developed a short-range lidar of high spatial resolution (Gaudfrin et al., 2019)(Gaudfrin et al., 2018b). The lidar inversion cannot be performed by means of the classical Klett-Fernald equation, because the reference layer used for the inversion is either impossible to access (horizontal lidar measurements, sky-to-ground lidar airborne measurements), or inaccessible because of finite lidar range. In the present paper, a modification of the conventional lidar equation is proposed in order to perform lidar inversions using a surface reference target (SRT) at relatively short range ($r_{max} \approx 100\,\mathrm{m}$). Precisely, a unified lidar equation for surface and volumetric scattering media is suggested, and it is then used for a new inversion equation, inspired from the Klett-Fernald equation, using a SRT.

Also a new technique to retrieve the lidar ratio without using any sunphotometer, Raman or radar measurements is presented and applied to an aerosol plume. This new inversion technique is both assessed theoretically and experimentally using real lidar measurements. A discussion and a conclusion follow and close the present paper.

## 2  Unified lidar equation for surface and volumetric scattering media

Currently, lidar inversion methods use a volumetric layer of the upper atmosphere (higher than $8\,\mathrm{km}$ of altitude above ground level) as a reference target. This volume is considered as being free of aerosols and made only of pure molecular constituents whose optical scattering properties are known. In our approach, we propose to use a SRT of known bidirectional reflectance distribution function (BRDF) $f_{r,\lambda}$ (in $\mathrm{sr}^{-1}$) (Nicodemus, 1965; Kavaya et al., 1983).

This requires to modify the usual lidar equation to make it suitable for both surface and volumetric targets.

For the single-scattering lidar equation, for which light has undergone only one scattering event, the measured backscattered power, at range $r$, can be written in a general way, *viz.* by considering both a surface target (Bufton, 1989; Hall and Ageno, 1970) and a volumetric target (Collis and Russell, 1976), as:

$$\mathcal{P}_\lambda(r, \theta_i) = \mathcal{P}_{p,\lambda} \frac{c\tau_\lambda}{2} \frac{A_{ef}}{r^2} \left\{ \beta_\lambda(r) + \frac{2}{c\tau_\lambda} f_{r,\lambda}(r_s, \theta_i) F_{cor} \right\} T_\lambda^2(r) \xi_\lambda(r) \eta_\lambda \tag{1}$$

where $\mathcal{P}_{p,\lambda}$ (in W) is the peak power of the laser source, $c \approx 3 \times 10^8\,\mathrm{m \cdot s^{-1}}$ the Einstein's constant, $\tau_\lambda$ (in s) the laser pulse duration (full width at half maximum), $A_{ef}$ (in $\mathrm{m}^2$) the telescope effective receiving area $\theta_i$ (in rad) the angle between the normal eigenvector to the SRT and the incident beam direction. It should be noted that in the particular case of a Lambertian surface $f_{r,\lambda}(r_s, \theta_i)$ can be easily expressed by spectral bidirectional reflectance factor $\rho_\lambda$ from $\rho_\lambda \cos\theta_i/\pi$ (Josset et al., 2018, 2010b; Haner et al., 1998). However, the general form of BRDF ($f_{r,\lambda}$) will be considered later in this work in order to not restrict the approach to specific cases. Also, the SRT is located at range $r_s$, $\xi_\lambda$ the dimensionless overlap function, $\eta_\lambda$ the dimensionless optical efficiency of the whole receiver. $\mathcal{P}_{p,\lambda}$ is a rectangular-shaped pulse in volumic lidar equation (Measures, 1992), *viz.* the ratio between the pulse energy and $\tau_\lambda$. In the case of lidar measurements on a SRT, the backscattered peak-power is not proportional to $\mathcal{P}_{p,\lambda}$. A corrective factor $F_{cor}$ depending on the real shape of the laser pulse is thus introduced. In the present case: $\mathcal{P}_{p,\lambda}^G = \mathcal{P}_{p,\lambda}^s F_{cor}$, with $\mathcal{P}_{p,\lambda}^G$ and $\mathcal{P}_{p,\lambda}^s$ the peak power of a Gaussian-shaped and a square laser pulse, respectively. Conservation of the pulse energy between these two kind of pulses gives $F_{cor} = 2(\ln 2/\pi)^{1/2}$ (Paschotta, 2008). The factor does not apply to the volume part of the lidar equation because, in this last part, the pulse profile is assumed to be constant over a rate duration $\tau_\lambda$. This approximation cannot be made on the backscatter peak of a SRT, because the backscattered energy is not integrated over a volume.

In Eq.1, $T_\lambda^2$ the back and forth atmospheric transmission throughout the environment between the lidar source and range $r$ (Swinehart, 1962):

$$T_\lambda(r) = \exp\left[ -\int_0^r \alpha_\lambda(x)\,\mathrm{d}x \right] \tag{2}$$

$\alpha_\lambda$ (in $\mathrm{m}^{-1}$) being the total extinction coefficient at wavelength $\lambda$, and range $r$: $\alpha_\lambda = \alpha_{b,\lambda} + \alpha_{a,\lambda}$. The subscripts "b" and "a" refer, respectively, to the contribution of the background (molecules, aerosols) and to the contribution of the aerosol volumetric target under investigation. The total backscattering coefficient $\beta$ (in $\mathrm{m}^{-1} \cdot \mathrm{sr}^{-1}$) is $\beta_\lambda = \beta_{b,\lambda} + \beta_{a,\lambda}$, with the same meaning as just above for the subscripts. By definition, the corresponding lidar ratios are $LR_{b,\lambda}(r) = \alpha_{b,\lambda}/\beta_{b,\lambda}$ and $LR_{a,\lambda}(r) = \alpha_{a,\lambda}/\beta_{a,\lambda}$, respectively.

The fundamental quantity measured by the lidar instrument is a voltage $V$ (in volts) which is proportionnal to the backscattered power: $V_\lambda(r) = R_{v,\lambda}\mathcal{P}_\lambda(r)$, where $R_{v,\lambda}$ is the detection constant (in $\mathrm{V \cdot W^{-1}}$) which determines the light-voltage conversion. It can be written using the instrumental constant: $C_{ins} = R_{v,\lambda}K_s$ (in $\mathrm{V \cdot m^3}$), where $K_s = \mathcal{P}_{p,\lambda} c\tau_\lambda A_{ef}\eta/2$. In the literature, $C_{ins}$ is obtained from $\mathcal{P}_\lambda$ while, herein, it comes from the voltage and therefore takes into account all the emission, collection, detection and acquisition chain.

In the sequel, for better readability, the subscript $\lambda$ and $\theta_i$ will not be written thereafter.

The range corrected lidar signal $V_\lambda(r)r^2$ is so:

$$S(r) = C_{ins}\left(\beta_a(r) + \beta_b(r) + f_r\frac{2}{c\tau}F_{cor}\right)\exp\left\{-2\int_0^r[\alpha_a(x) + \alpha_b(x)]\,\mathrm{d}x\right\} \tag{3}$$

To remove the $\alpha-$dependence in the exponential term, we will replace $\alpha_a$ and $\alpha_b$ by $LR_a$ and $LR_b$, respectively, and introduce the term:

$$LR_a(r)\exp\left\{-2\int_0^r\beta_b(x)[LR_a(x) - LR_b(x)]\,\mathrm{d}x\right\} \tag{4}$$

as detailed in Ansmann and Müller (2004). With such modifications, the final lidar equation for surface and volumetric scatterers can thus be written as:

$$S(r)LR_a(r)\exp\left\{-2\int_0^r\beta_b(x)[LR_a(x) - LR_b(x)]\,\mathrm{d}x\right\} = C_{ins}\left[Y(r) + LR_a(r)\frac{2f_r}{c\tau}F_{cor}\right]\exp\left[-2\int_0^r Y(x)\,\mathrm{d}x\right] \tag{5}$$

with $Y(r) = LR_a(r)\left[\beta_b(r) + \beta_a(r)\right]$.

Thereafter, in order to highlight the expression to solve, it is convenient to define background corrected transmission factor:

$$D(0,r) = \exp\left\{-2\int_0^r\beta_b(x)\left[LR_a(x) - LR_b(x)\right]\,\mathrm{d}x\right\} \tag{6}$$

and $W(r) = S(r)LR_a(r)D(r)$. Finally, Eq. 3 becomes:

$$W(r) = C_{ins}\left[Y(r) + LR_a(r)\frac{2f_r}{c\tau}F_{cor}\right]\exp\left[-2\int_0^r Y(x)\,\mathrm{d}x\right] \tag{7}$$

We will now introduce the lidar framework adapted to the radiative parameter retrieval of a volumetric scattering medium with a known SRT.

## 3   New lidar inversion technique

### 3.1   Radiative parameters identification

The current Klett-Fernald inversion method consists in determining $C_{ins}$ using the high atmosphere as a reference and to fix the $LR_a$ *a priori* . In this paper, $C_{ins}$ is determined by means of a SRT located at range $r_s$. So:

$$C_{ins} = \frac{c\tau}{2f_r F_{cor}}W(r_s)\frac{1}{LR_a(r_s)}\exp\left[2\int_0^{r_s}Y(x)\,\mathrm{d}x\right] \tag{8}$$

It is worth mentioning that $LR_a(r_s)$ is the lidar ratio just before the SRT and $Y(r_s) = 0$ (only at the SRT). Also, obviously, for $r < r_s$, $f_r = 0$. Inserting Eq. 8 in Eq. 7 gives:

$$W(r) = \frac{c\tau}{2 f_r F_{cor}} \frac{W(r_s)}{LR_a(r_s)} Y(r) \exp \left[ 2 \int_r^{r_s} Y(x) \, \mathrm{d}x \right]$$
(9)

This equation applies only before the SRT and can be solved by integrating both sides from $r$ to $r_s$ (Vande Hey, 2014). The exponential term is (see Appendix):

$$\exp \left[ 2 \int_r^{r_s} Y(x) \, \mathrm{d}x \right] = 1 + \frac{4 f_r F_{cor} LR_a(r_s)}{c\tau W(r_s)} \int_r^{r_s} W(x) \, \mathrm{d}x$$
(10)

Plugging Eq. 10 into Eq. 9, we obtain in the following:

$$Y(r) = W(r) \left[ \frac{c\tau W(r_s)}{2 f_r F_{cor} LR_a(r_s)} + 2 \int_r^{r_s} W(x) \, \mathrm{d}x \right]^{-1}$$
(11)

Using the definitions of $Y(r)$ and $W(r)$ (see above), $\beta_a(r)$ can be written as:

$$\beta_a(r) = S(r) D(0, r) \left[ \frac{c\tau S(r_s) D(0, r_s)}{2 f_r F_{cor}} + 2 \int_r^{r_s} S(x) LR_a(x) D(0, x) \, \mathrm{d}x \right]^{-1} - \beta_b(r)$$
(12)

Multiplying the numerator and the denominator of the first term on the right-hand side of the subtraction by $D(r_s, 0)$, this expression becomes:

$$\beta_a(r) = S(r) D(r_s, r) \left[ \frac{c\tau S(r_s)}{2 f_r F_{cor}} + 2 \int_r^{r_s} S(x) LR_a(x) D(r_s, x) \, \mathrm{d}x \right]^{-1} - \beta_b(r)$$
(13)

Then, by definition of the lidar ratio, we deduct $\alpha_a(r) = LR_a(r) \beta_a(r)$. Eq. 13 is similar to the one defined by Klett (1981), except that $\beta_b$ in Eq. 13 contains also the contribution of the aerosol background.

Assuming that the properties of the SRT are well known, the most critical parameter is $LR_a(r)$. Giving a value for $LR_a$ requires an *a priori* knowledge of the volumetric target under study whereas the main objective of lidar remote sensing is precisely to characterize the medium investigated. *A prioris* are always topic of discussions and are more or less severe flaws in lidar measurements.

Equation 13 can also be applied on the important context of airborne observations. In this case, it is necessary to know the ground $BRDF$.


### 3.2  Determination of $LR_a$ and $\beta_a$: methodology

The objective is to retrieve first $\beta_a(r)$ and $LR_a$ (and then to deduce $\alpha_a(r)$) without any *a priori* about the medium considered. Two lidar measurements are performed: the first one (signal $V_s$) in the absence of the volumetric aerosol medium of interest

and a subsequent one (signal $V_{sv}$) in its presence. The SRT is obviously present for both measurements. The two measurements should be performed close in time in order to avoid that the background environment evolves too much. The experimental setup of these lidar measurements is illustrated on Fig. 1

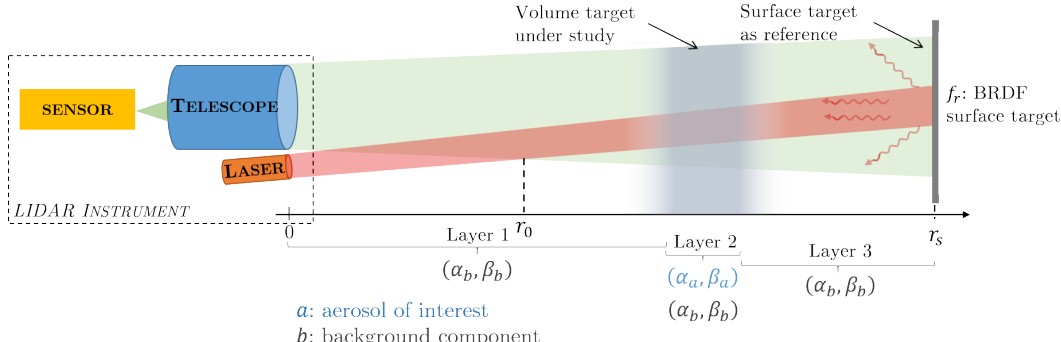

**Figure 1.** Illustration of the experimental setup

By definition, the half-logarithmic ratio of $S_s$ and $S_{sv}$ corresponds to the total extinction of the volumetric media under study: $\alpha_{tot} = \ln[S_s(r_s)/S_{sv}(r_s)]/2$. Using $S_s$, $C_{ins}$ can be determined independently of the volumetric medium of interest:

$$C_{ins} = \frac{c\tau}{2 f_r F_{cor}} S_s(r_s) \exp\left[ 2 \int_0^{r_s} \alpha_b(x)\,\mathrm{d}x \right] \tag{14}$$

which is Eq. 8 with $\alpha_a = 0$. $C_{ins}$ and $\alpha_{tot}$ are used in objective functions to retrieve $LR_a$, assumed to be uniform – $r-$independent. The first objective function is:

$$\varepsilon_1 = \left| \int_{r_0}^{r_s} \alpha_a(x)\,\mathrm{d}x - \alpha_{tot} \right| \tag{15}$$

where $\alpha_a$ is the retrieved profile of extinction using Eq. 13 and $LR_a$. The medium is assumed to be at range of full overlap ($r > r_0$), so that $\alpha_{tot}$ must correspond to the integrated extinction. A second objective function:

$$\varepsilon_2 = \left| \int_{r_0}^{r_s} [S_{sv}(x) - S_{sim}(x)]\,\mathrm{d}x \right| \tag{16}$$

is introduced in order to minimize the difference between $S_{sv}$ and the simulated signal $S_{sim}$ obtained from the retrieved $\beta_a$ and $\alpha_a$ and from $C_{ins}$.

The methodology is presented on Fig. 2. The molecular background contribution is computed from pressure and temperature data as in Bucholtz (1995), while the aerosol background contribution is estimated by means of radiative transfer codes, *e.g.* MATISSE (Simoneau et al., 2002; Labarre et al., 2010) or MODTRAN (Berk et al., 2008, 2014). Another solution consists in

using a realistic value of the visibility $\mathcal{V}$ (in $\text{km}^{-1}$) and the Koschmieder's relation (Horvath, 1971; Elias et al., 2009; Hyslop, 2009) at $550\,\text{nm}$ (maximum human eye sensitivity): $\mathcal{V}\alpha_b \approx 3.9$

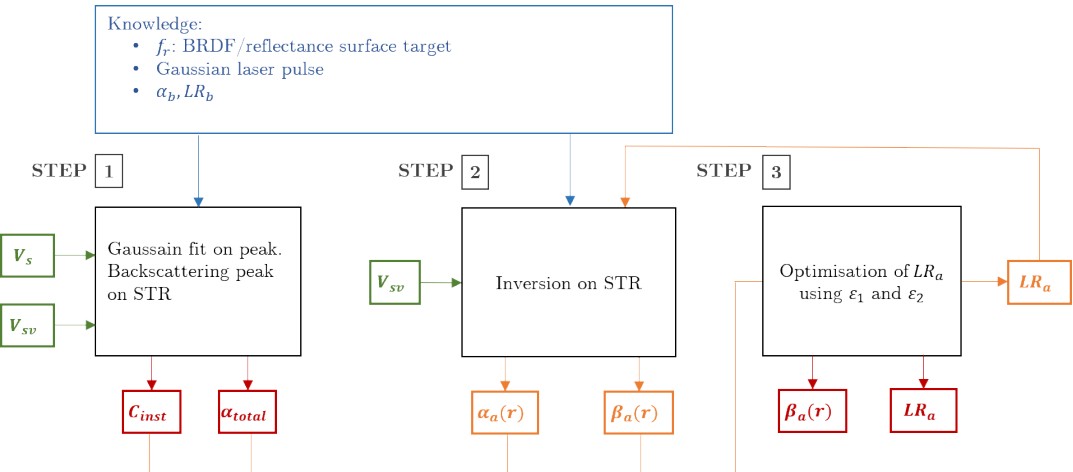

**Figure 2.** Diagram illustrating how the inversion algorithm allows to retrieve the $\beta_a(r)$ and $LR_a$ without assumptions on the volume medium of interest. In green the lidar signals inputs, in orange the intermediate calculations during the optimization procedure and in red the code outputs.

The signals $V_s$ and $V_{sv}$ are introduced in the inversion procedure, which is organized around three main steps (Fig. 2):

1. A Gaussian fit is first applied on the backscattered signal from the SRT, *i.e.* $V_s(r_s)$ and $V_{sv}(r_s)$, that gives the amplitude of the backscattering, the position of this peak and its width in position. From these gaussian models, one can obtain $\alpha_{tot}$ (from its definition, see above) and $C_{ins}$ from Eq. 14. Note: When the target is tilted with respect to the lidar-target line, the backscatter peak of surface target will not be symmetrical. An other fit should be used as a log-normal function.

2. A first lidar inversion is realized using Eq. 13 with $LR_a = 50\,\text{sr}$ at the beginning of the inversion procedure. This value has been chosen because it corresponds to the average $LR_a$ data of the literature. For that, the gaussian model $V_{sv}$ obtained at step 1 is used for signal $S(r_s)$ in Eq. 13. A first range-profile $\beta_a(r)$ is thus obtained at the end of this second step.

3. The above $\beta_a(r)$ and $LR_a$ allow to determine $\alpha_a(r)$ whose $r-$integration is then compared with $\alpha_{tot}$ in the minimization procedure of Eq. 15. At each iteration, the $LR_a$ is modified in order to reduce $\varepsilon_1$. The new $\beta_a(r)$, $LR_a$, and so $\alpha_a(r)$ are then used to compute a simulated lidar signal $S_{sim}$ whose comparison with $S_{sv}$ is minimized according to Eq. 16. In this algorithm, the iterative procedure ends up when $\varepsilon_1 + \varepsilon_2 \leq 10^{-6}$ is reached. A number of 19 iterations is generally enough, depending on the first value of $LR_a$ introduced initially (step 2). At the end of this step, one thus obtains final $\beta_a(r)$, $\alpha_a(r)$ and $LR_a$. The minimization procedure used is the one implemented by Kraft (1988). Eq. 15 is the most important since it determines the rapiditiy of convergence. Eq. 16 is helpful but not critical.

## 4    Theoretical behavior of the retrieval procedure

### 4.1    Theoretical lidar signals

The inversion method described above is tested using theoretical lidar signals generated by PERFALIS [1] (Gaudfrin et al., 2018a). As summarized in Table 1, the simulated atmosphere is composed of three layers and of a SRT of BRDF $f_r = 0.20/\pi$ located at $r_s = 100\,\mathrm{m}$. Pressure and temperature are uniform ($1040\,\mathrm{hPa}$ and $290\,\mathrm{K}$) and the continental aerosol background is chosen so that it corresponds to $\mathcal{V} = 47\,\mathrm{km}$ (Hess et al., 1998). In addition, $\beta_b = 9.97 \times 10^{-6}\,\mathrm{m}^{-1} \cdot \mathrm{sr}^{-1}$ and $LR_b = 118.56\,\mathrm{sr}$. The signal $V_s$ is generated from the background components and the SRT, while the signal $V_{sv}$ is generated considering an aerosol plume aerosol between $20 - 30\,\mathrm{m}$ (second layer). The plume backscatter coefficient is $\beta_a = 7.14 \times 10^{-5}\,\mathrm{m}^{-1} \cdot \mathrm{sr}^{-1}$ and $LR_a = 70\,\mathrm{sr}$. Multiple scattering is assumed to be negligible. For dense atmosphere and wider field of view, Eq. 1 has to be corrected by an appropriate factor (Bissonnette, 1996) in order to consider higher orders of scattering events.

|  | Notation | Layer 1 | Layer 2 | Layer 3 | SRT ($f_r = 0.20/\pi\ \mathrm{sr}^{-1}$) |
|---|---|---|---|---|---|
| Range | $r$ (in m) | $0 - 20$ | $20 - 30$ | $30 - 100$ | 100 |
| Background components | $\alpha_b$ (in $\mathrm{m}^{-1}$) | $1.18 \times 10^{-3}$ | $1.18 \times 10^{-3}$ | $1.18 \times 10^{-3}$ |  |
|  | $\beta_b$ (in $\mathrm{m}^{-1} \cdot \mathrm{sr}^{-1}$) | $9.97 \times 10^{-6}$ | $9.97 \times 10^{-6}$ | $9.97 \times 10^{-6}$ | X |
|  | $LR_b$ (in sr) | 118.56 | 118.56 | 118.56 |  |
| Volumetric medium | $\alpha_a$ (in $\mathrm{m}^{-1}$) |  | $5.00 \times 10^{-3}$ |  |  |
|  | $\beta_a$ (in $\mathrm{m}^{-1} \cdot \mathrm{sr}^{-1}$) | X | $7.14 \times 10^{-5}$ | X | X |
|  | $LR_a$ (in sr) |  | 70 |  |  |

**Table 1.** Input optical parameters of the scene used in the lidar simulator (PERFALIS code) as illustrated on Fig. 1

Inversion methods are generally applied to averaged signals in order to increase the $SNR$. In lidar remote sensing, the noise can be, approximately, considered as a white Gaussian noise (Li et al., 2012; Mao et al., 2013; Sun, 2018). In order to assess the impact of noise in the inversion method (see Section 3), a Gaussian noise of null mean value and a standard deviation of $1.5 \times 10^{-5}\,\mathrm{a.u.}$ is introduced in the theoretical lidar signals. Figure 3 displays the theoretical noised signals $V_s$ and $V_{sv}$. As expected, because of light extinction by the plume, $V_{sv}(r_s)$ is lower than $V_s(r_s)$ by 9%. Four datasets are then generated, with respectively, an averaging over 20, 50, 100, and 200 signals, from $V_s$ and $V_{sv}$, and, in addition, a fifth signal without noise is considered (Fig. 4).

---

[1]PERFormence Assesment for LIdar Systems

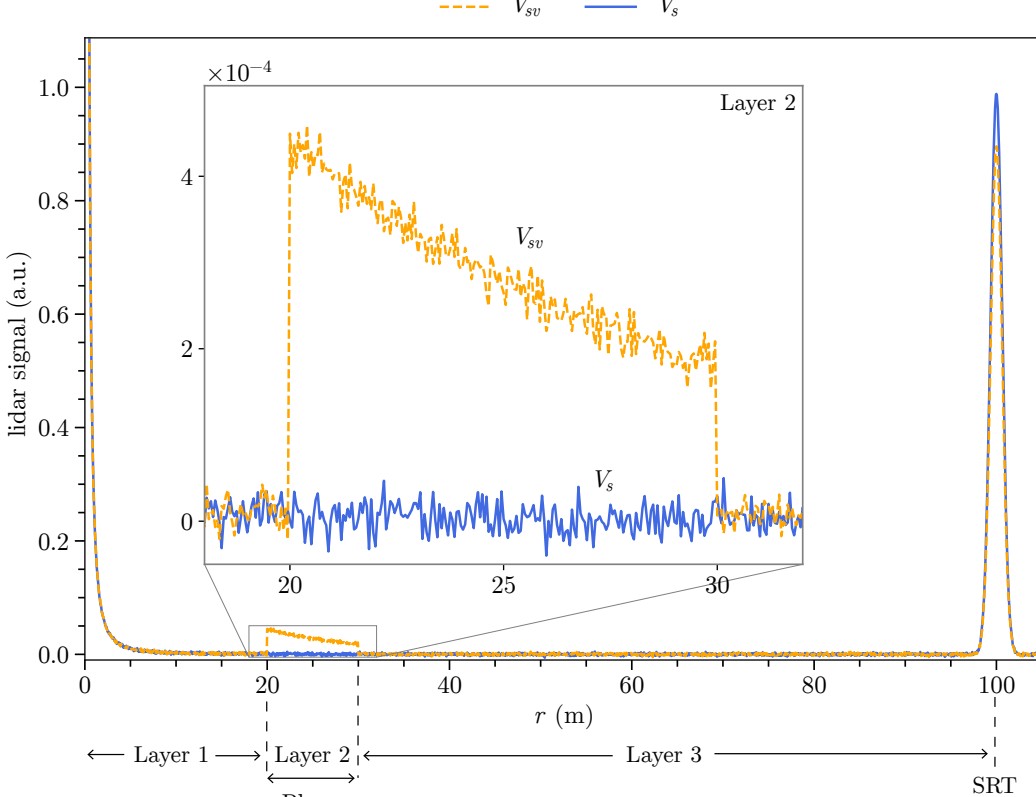

**Figure 3.** Theoretical noised lidar signals from a SRT $V_s$ (blue line) and in the presence of an aerosol plume $V_{sv}$ (orange dashed line). Simulations have been performed at $532\,\text{nm}$ with molecular and continental aerosol background contributions.

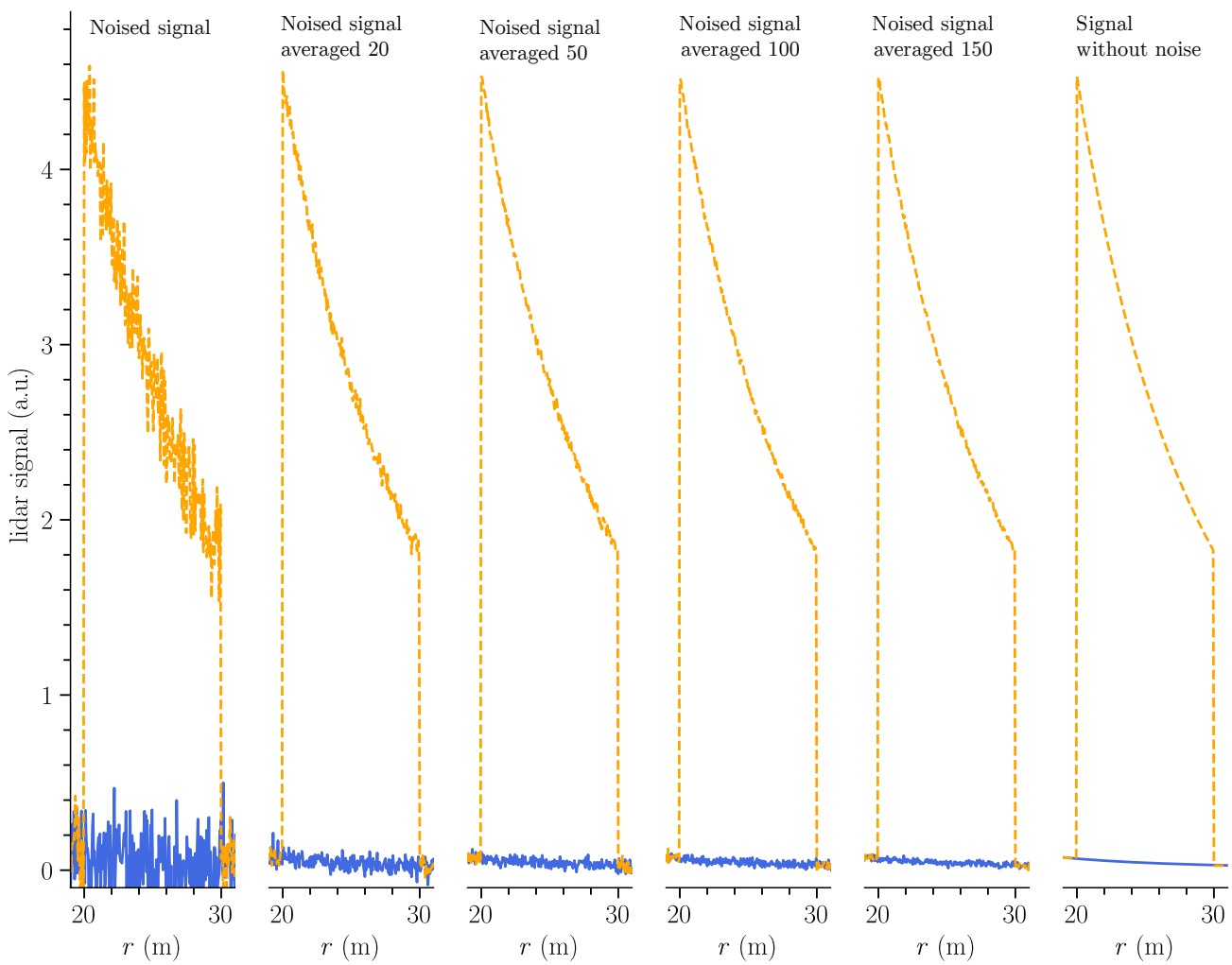

**Figure 4.** Lidar data sets used in the inversion method. In blue, (orange) the lidar signal in the absence (presence) of the volumetric media under study.

## 4.2 Noise impact on $\beta_a$ and $LR_a$ retrievals

$LR_a$ is retrieved using Eq. 15. In addition to the six lidar-dataset described above, four different conditions of inversion are considered. In condition 1 the exact data of the background components are used as an input of the inversion algorithm. For conditions 2 and 3, $\beta_b$ is over- and underestimated by $20\%$ compared to the data used to generate the theoretical signals. In conditions 1 to 3, the inversion technique is performed over the entire signal range. Condition 4 is the same as condition 1, but the aerosol plume is spatially delimited. Table 2 summarizes the four-conditions for the six datasets. It is worth mentioning that


noised lidar signals obviously results in noised retrieved $\beta_a(r)$. Thus, to quantify the performance of the inversion technique, we consider the average value $\overline{\beta_a}$ of the plume. The retrieved value of $LR_a$ can be directly compared to the theoretical value.

| Conditions | 1 | 2 | 3 | 4 |
|---|---|---|---|---|
| Exact background constituents | X | | | X |
| $\beta_b + 20\%$ | | X | | |
| $\beta_b - 20\%$ | | | X | |
| Spatially bounded plume | | | | X |

**Table 2.** Conditions on the optical properties of the background components for the inversion method.

Figure 5 displays $\overline{\beta_a}$ for the six datasets and the four inversion conditions. It varies from $7.11 \times 10^{-5}$ to $7.22 \times 10^{-5}\,\mathrm{m}^{-1} \cdot \mathrm{sr}^{-1}$,

which means an error of approximately $1\%$ in comparison to the theoretical value. Conditions 2 and 3 result in a translation of the corresponding curve of $\pm 0.4\%$ with respect to the curve associated to condition 1, because of the over- and underestimation of $20\%$ introduced in $\beta_b$. The performance is better for condition 4 whatever the dataset, since the maximum error is $0.5\%$ for noised signals. The spatially bounded aerosol layer is often applied in inversion methods, and seem to herein improve the inversion method.

For signal lidar whithout noise, $\overline{\beta}_a$ is not exactly equal to the theoretical value, maybe because of numerical computation errors in the inversion algorithm. Such a numerical error is about $0.12\%$ (condition 1) and $0.04\%$ (condition 4).

Fig. 6 is similar as Fig. 5 but considering $LR_a$. One obtains values ranging from 66 to $74\,\mathrm{sr}$, with a maximum error of $5\%$ compared to the theoretical value. In conditions 1, 2, and 3, using averaged noised signals has no consequence on the retrieved value of $LR_a$, contrary to what was obtained for $\overline{\beta}_a$.

In condition 1, the maximum error is $2.1\%$. The graphs corresponding to conditions 2 and 3 are translated, with respect to the graph under to condition 1, by about $\pm 3\%$, and permuted respectively to the same but for $\overline{\beta}_a$. Nevertheless, the errors remain low with a maximum of $5\%$ (condition 2) if 50 signals are averaged. However, under condition 4, the $LR_a$ is much better for averaged signals and remains quite good for noisy signal (not averaged) with an error rate of $0.6\%$. Again, it seems that the spatial limitation of the plume increases the accuracy of the retrieval $LR_a$. Condition 1 remains however efficient for noised

signals since deviation is below $2.1\%$. In the case of lidar signal whithout noise, the retrieved $LR_a$ are not exactly equal to the theoretical $LR_a$; numerical computation errors are about $0.13\%$ (condition 1) and $0.05\%$ (condition 4). An error of $\pm 20\%$ on $\beta_b$ introduced initially will result in an under- or overestimation $LR_a$ by $\pm 3\%$. Condition 4 is preferable to retrieve $LR_a$.

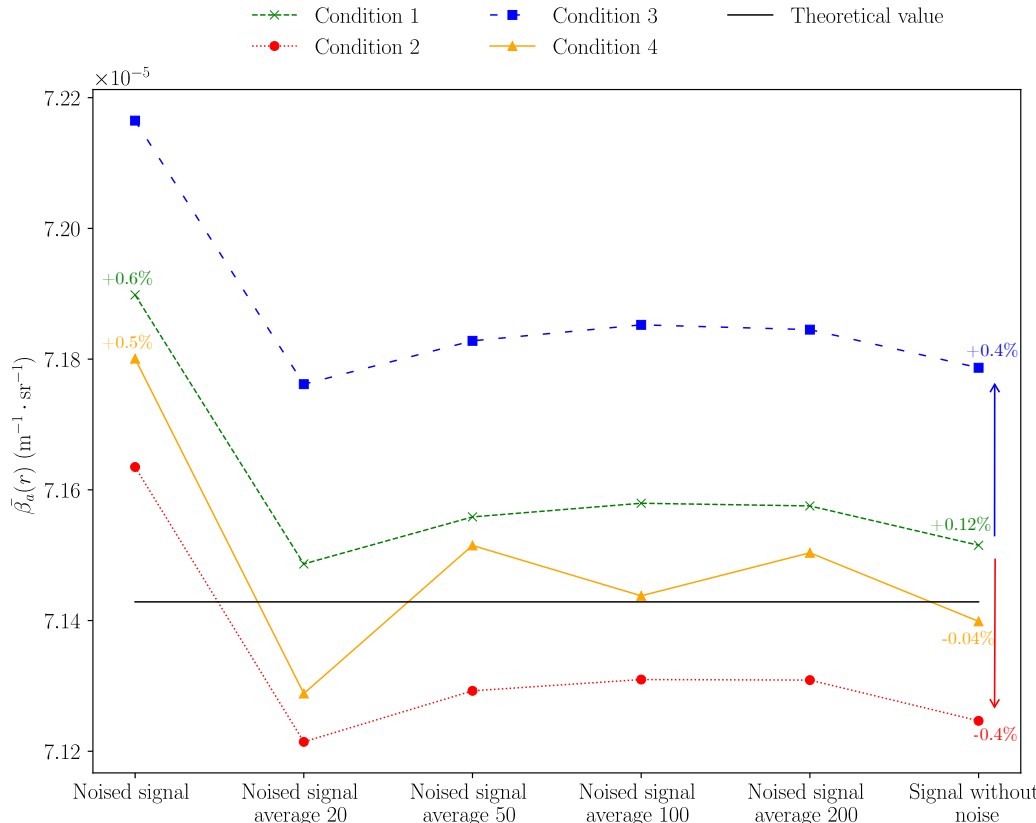

**Figure 5.** Retrieved $\overline{\beta}_a$ for six datasets and four different conditions of inversion.

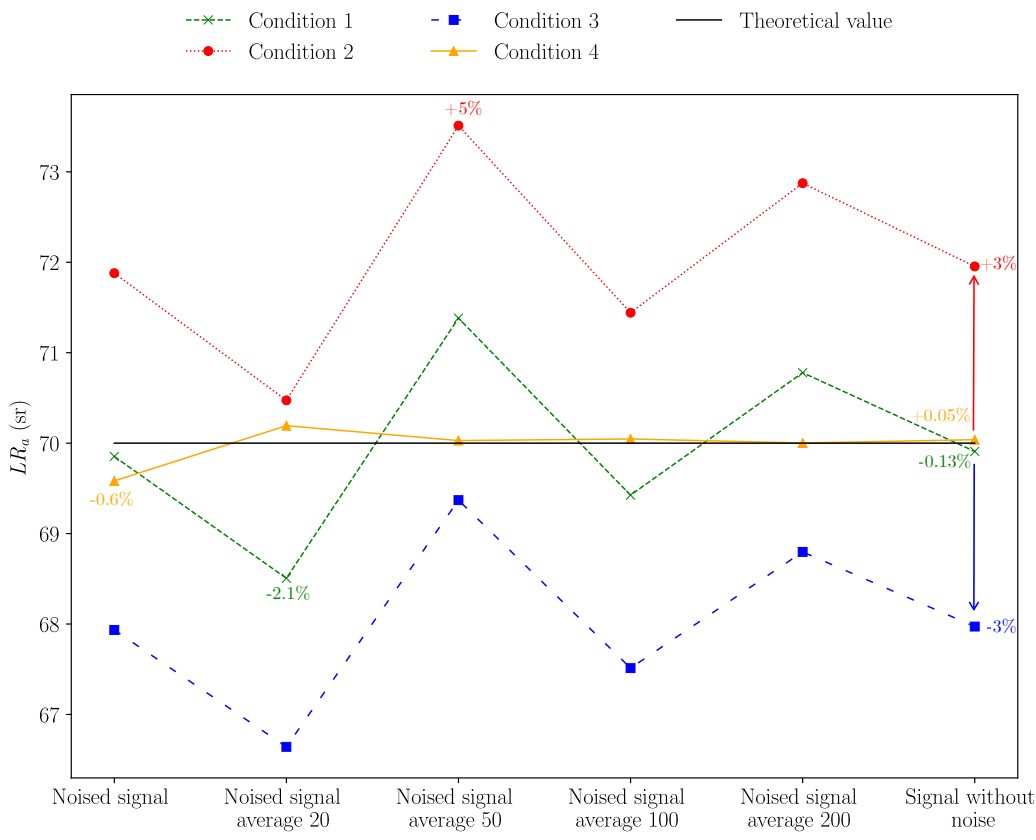

**Figure 6.** Retrieved $LR_a$ for six datasets and four different conditions of inversion.

Note that the formalism and methodology adopted here to retrieve the lidar ratio are efficient as long as the peak backscattering of the SRT is present on the lidar signal. The method has been evaluated, in this manuscript, for short range around $100\,\mathrm{m}$ because our research focus on application at this range. However, the algorithm developed does not present any limit with respect to the range provided that measurements are made below $1\,\mathrm{km}$ of range (this value depends of the power of laser sources) with respect to our applications. However, at first sight there is no limit to the application of the method to measurements at longer ranges such as more than $1\,\mathrm{km}$ measurements.

### 4.3 Plume optical property retrieval

The above study allowed us to test the new inversion method on noised signals, for different conditions of inversion, as a function of the number of signals averaged. Thereafter, lidar inversion is performed considering a spatially bounded plume and 100 signals for averaging. This last condition has been chosen because it corresponds to the number of signals available in less than $0.1\,\mathrm{s}$ with our lidar system (see Section 5). The theoretical results obtained by the inversion method with 100 averaged signals is also quite good (see above). Figure 9 displays the retrieved $\beta_a$ if a theoretical lidar signal is introduced as a first

guess. Table 3 lists the retrieved $\overline{\beta}_a$ and $LR_a$. Compared to theoretical values, errors are less than $0.7\%$ for $LR_a$ and below

$0.1\%$ for $\overline{\beta}_a$, although a peak of $2.2\%$ is observed at $r = 28.8\,\mathrm{m}$.

|  | $LR_a$ | $\overline{\beta}_a$ |
|---|---|---|
| Value | $70.05\,\mathrm{sr}$ | $7.14 \times 10^{-5}\,\mathrm{m}^{-1} \cdot \mathrm{sr}^{-1}$ |
| Error | $0.07\%$ | $0.01\%$ |

**Table 3.** Plume optical property retrieved and associated errors

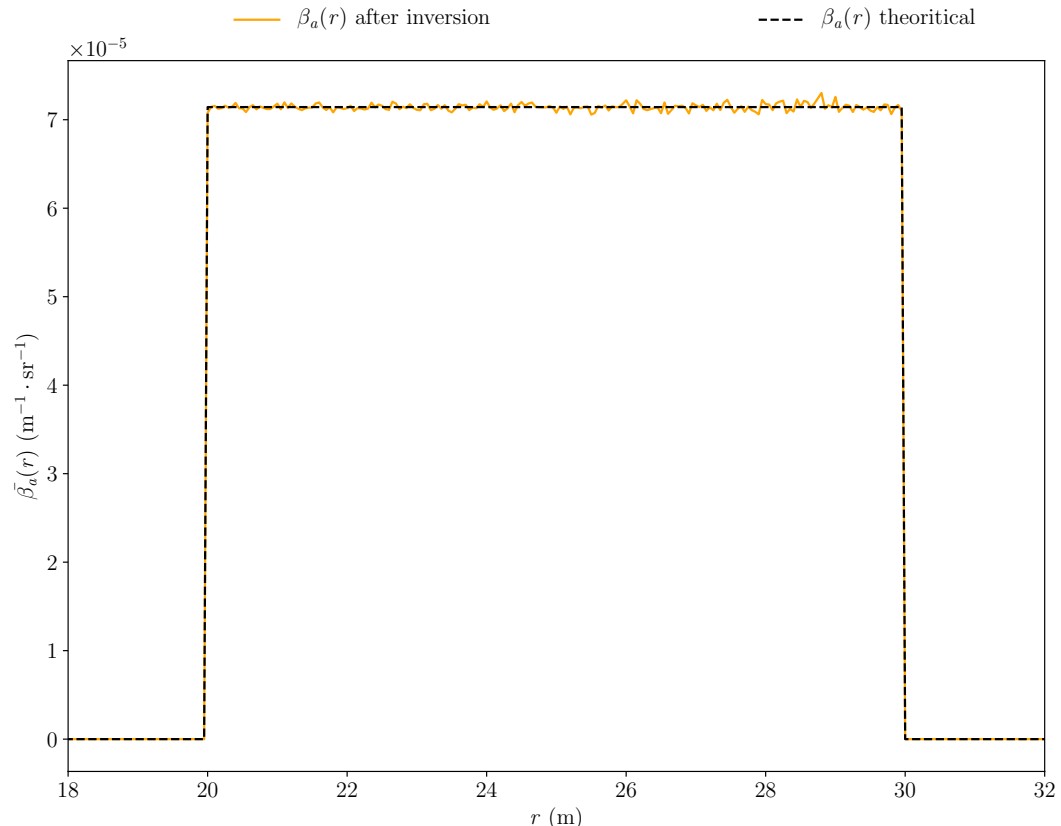

**Figure 7.** Retrieved $\beta_a(r)$ (orange solid line) for 100 theoretical averaged lidar signals and initial $\beta_a(r)$ (dark dashed line).

## 5   Case of real measurements

Our new inversion technique is now applied to real lidar measurements. The instrument used is named COLIBRIS [2] (Gaudfrin et al., 2018b)(Ceolato and Gaudfrin, 2018). This lidar is able to perform short-range measurements $(r_0 < 5\,\mathrm{m})$ at high spatial

---

[2] Compact lidar for Broadbord polaRIsation Spectral multi-Static measurement

resolution (lower than $0.25\,\mathrm{m}$). A Nd:YAG microchip laser source of the HORUS-LEUKOS company is used with a pulse energy peaking at $532\,\mathrm{nm}$ of $7.3\,\mu\mathrm{J}$ and a repetition rate of $1\,\mathrm{kHz}$. The backscattered light is collected by a Cassegrain telescope. In the detection part, a dichroic filter for the elastic channel is used before a photomultiplier tube. The signal is digitized at a sample frequency of $3\,\mathrm{GHz}$ after been amplified.

## 5.1 Description of the experimental operations

The lidar measurements are performed horizontally as illustrated in Fig. 8. A Lambertian Zenithal SRT (SphereOptic) with a $f_r = 0.20/\pi$ is placed at $52\,\mathrm{m}$ far away from the source. Its spectral bidirectional reflectance has been checked using laboratory bench measurements (Ceolato et al., 2012). The mean direction of the laser beam is parallel to the normal of the surface.

The repetition laser source has repetition frequency of $1\,\mathrm{kHz}$. In order to increase the $SNR$, we preprocess the measurements from three lidar measurements:

– Signal 1. The first measurement is made by occulting the emitted laser beam to get a measure of the background scene (contribution of passive illumination);

– Signal 2. The second measurement is made by occulting the telescope to estimate the dark noise of the instrument;

– Signal 3. The last measurement is made without any occultation.

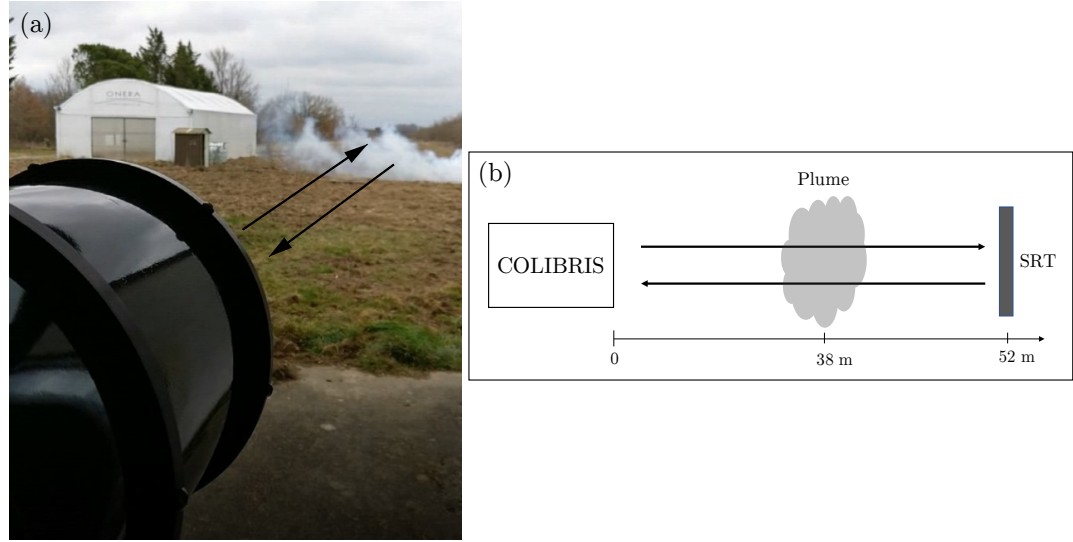

**Figure 8.** Experimental setup in an horizontal configuration. A fog-oil plume is generated between the lidar and the Lambertian SRT. (a) Photo and (b) illustration of the experimental setup with fog-oil plume.

For a given acquisition period, these three series of measured signals are averaged. The averaged signals of the background
radiation and the dark noise (signals 1 and 2) are then subtracted from the signal 3 such as : signal 3 - (signal 1 - signal 2).

Fig. 9 shows the lidar results on a volume/surface target over a period of $2\,\text{s}$. This corresponds to $2\,000$ signals per serie of measurements. During this period, we assume that the environment does not evolve significantly. The curves $V_{sv}$ and $V_s$ are the measurements in the presence and in the absence of oil smoke with SRT, respectively. The oil plume signal is visible between $37.5\,\text{m}$ and $41\,\text{m}$.

The high-speed sampling allows a measurement every $\text{cm}$ along the line of sight. Combined with a short pulse duration of the laser source ($1.7$ ns), this makes it possible to highlight local variations concentration in the order of $25$ cm inside the plume with the presence of two maxima at $38\,\text{m}$ and $39\,\text{m}$ from lidar. The peak of backscattering of the SRT is also well sampled. The signal amplitude corresponding to the backscatter of the SRT is lower on $V_{sv}$ than on $V_s$ because of the presence of the oil plume.

During measurement, the pressure, temperature and visibility are respectively $1016\,\text{hPa}$, $288\,\text{K}$ and $30\,\text{km}$. These data are used to compute $\beta_b$ as described in Section 4.

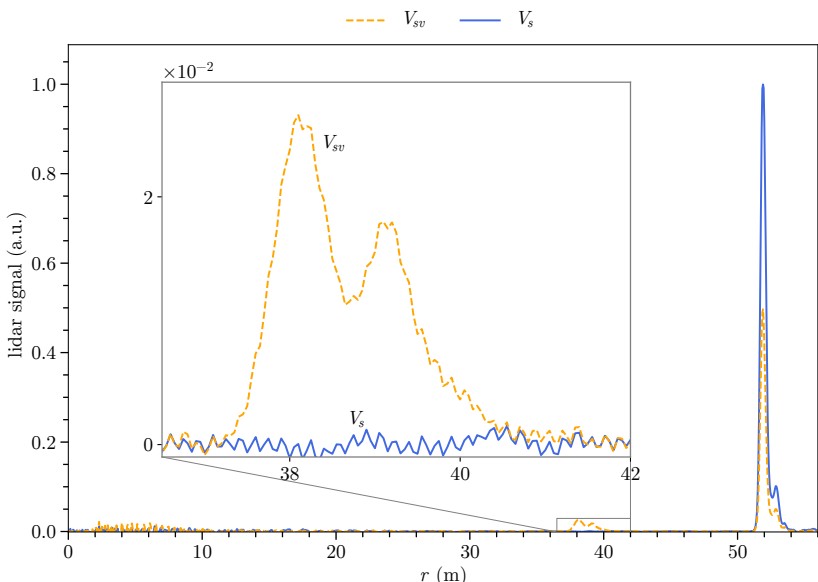

**Figure 9.** Lidar measurements of the experimental setup. In blue, (orange) the lidar signal in the absence (presence) of the oil-fog plume under study.

## 5.2   Optical property retrieval: fog-oil plume

The signals $V_s$ and $V_{sv}$ are used in the inversion procedure as described in Sections 3 and 4. The plume is spatially bounded (condition 4).

The retrieved $\beta_a(r)$ is displayed in Fig. 10. In the densest range of the plume $\beta_a \approx 2 \times 10^{-3}\,\mathrm{m^{-1}\,sr^{-1}}$. Also, the retrieved $LR_a$ is around $98\,\mathrm{sr}$. According to Bohlmann et al. (2018), this value corresponds, as expected, to smoke particles (at $532\,\mathrm{nm}$, the lidar ratio ranges from $80$ to $100\,\mathrm{sr}$).The optical properties of the oil-fog plume of experimental retrieved with inverse method are summarized in Section 4.

    The lidar signal reproduced from the retrieved $\beta_a(r)$, $LR_a$ and of the instrumental constant deduced from the Eq. 14 gives
a standard deviation from the exact value of $1.5 \times 10^{-5}\,\mathrm{a.u.}$ This shows the consistency and reliability of the new inversion method proposed in this paper.

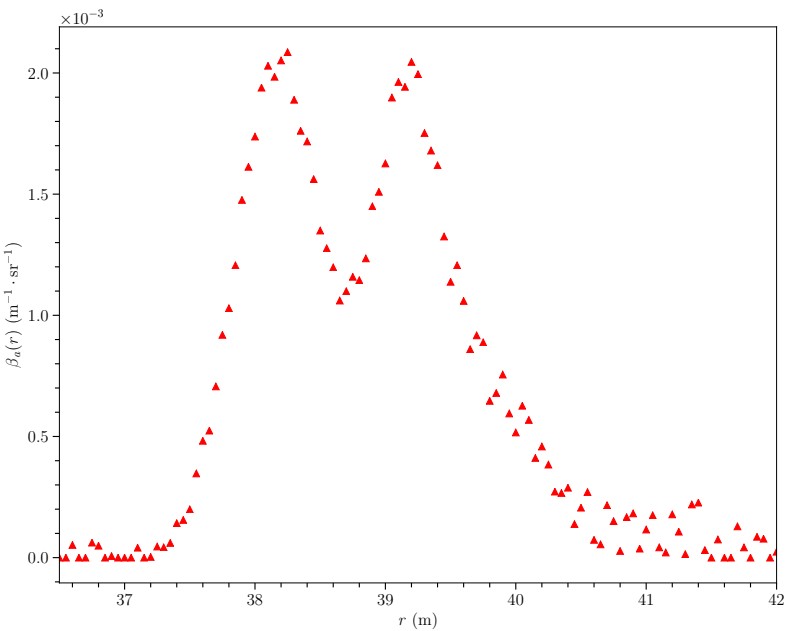

**Figure 10.** Retrieved $\beta_a(r)$ for real measurements with $LR_a = 98\,\mathrm{sr}$.

| $LR_a$ (sr) | 98 |
|:---:|:---:|
| $\beta_{a,max}$ (m$^{-1}\cdot$sr$^{-1}$) | $2.1 \times 10^{-3}$ |
| $\alpha_{a,max}$ (m$^{-1}$) | $2.1 \times 10^{-1}$ |
| Optical thickness | $3.6 \times 10^{-1}$ |

**Table 4.** Optical properties of oil-fog plume in experimental setup at $532\,\mathrm{nm}$

## 6   Conclusions

In this paper, a new method has been introduced for lidar measurement inversion in a situation for which a volumetric layer (molecular Rayleigh scatterers) of the high troposphere is not available (*e.g.* airborne lidar observations, horizontal config-

uration of measurements). This method is based on a new expression of the lidar equation which allows us to use a surface reference target of a known BRDF instead of a volumetric one. This new formalism permits to invert short-range lidar measurements for which conventional inversion techniques can not directly be applied. Similarly to common inversion techniques, our method requires to introduce a background component (molecular and particulate contributions) that can be estimated either from radiative models or deducted from measurements of temperature, pressure, and visibility conditions.

Also, a new algorithm has been developed to retrieve, without any *a priori* assumptions relative to the medium to be characterized (aerosol plume), the backscattering coefficient ($\beta_a$) and lidar ratio ($LR_a$) of an aerosol plume, between the lidar and the surface target reference. In other words, our technique method does not need to introduce any lidar ratio as an input for our inverse algorithm. For that, two lidar measurements are necessary: with and without the aerosol plume under consideration.

Comparing these two signals, one can retrieve the total extinction coefficient of the medium analysed and the instrumental constant of the lidar instrument. These two informations are used to constrain the inversion algorithm and finally to identify $LR_a$.

This algorithm has been first investigated using theoretical (simulated) lidar signals. The quality of the retrieval has been assessed by introducing noise in the simulated signals and by considering various conditions of inversion differing, in particular, from one another according to the initial error introduced in the backscattering coefficient of the aerosol background. Thus, the robustness of algorithm has been shown, since in all the cases, the error on the retrieved values (*viz.* in $\beta_a$ and $LR_a$) is less than $5\%$, at most. Also, we have found that inversion is better for spatially bounded aerosol plume.

The inversion algorithm has then been applied on real lidar short-range measurements of an oil-fog plume. The retrieved $\beta_a$ and $LR_a$ of the plume agree with values found in the literature for smoke-like particles. Moreover, thanks to the determination of the instrumental constant, the measured signal has been computed from the inverted products, and an absolute error of $10^{-5}$ a.u. between the measure and the post-processed simulation has been encountered.

However, it is worth mentioning that the method proposed herein to find $LR_a$ has some limitations. Precisely, the sensitivity of the lidar must be sufficient to detect the signal of weakly thick or weakly backscattering plume. Indeed, since measurements are performed in the absence and in the presence of the medium, by means of a hard surface target of reference of known reflectance, the algorithm converges less easily for very weakly diffusing plumes.

The new inversion technique presented in this paper suggests new airborne lidar applications operated at low altitude from aircraft (helicopters, airplanes), but requires *a priori* knowledge of the reflectance of the SRT.

Even if some models exist for the BRDF of surfaces (Bréon et al., 2002; Lobell and Asner, 2002; Mishchenko et al., 1999), their use seems difficult to implement because of the diversity of encountered surfaces during airborne measurements. Nevertheless, it may be possible to identify the reflectance of the ground surface by means of a spectroradiometer imager (Poutier et al., 2002; Miesch et al., 2005; Josset et al., 2018). The combination of these measurements with the herein proposed inversion method would be *a priori* be complementary to establish new methods of calibration for downlooking lidar measurements (spaceborne or airborne lidars). The evaluation of the method proposed in this paper, considering the uncertainty of the target reflectance, has not been performed. It will be the topic of future works.

## 7 Appendix

To solve Eq. 9, the exponential term can be written under another form. The method proposed by Vande Hey (2014) consists in integrating both members of the equation from $r$ to $r_s$. So:

$$\int_r^{r_s} W(x)\,\mathrm{d}x = \frac{c\tau}{2 f_r F_{cor}} \frac{W(r_s)}{LR_a(r_s)} \int_r^{r_s} \left\{ Y(x) \exp\left[ 2 \int_x^{r_s} Y(r)\,\mathrm{d}r \right] \right\}\,\mathrm{d}x$$

Since:

$$\frac{\mathrm{d}}{\mathrm{d}x}\left\{ \exp\left[ 2\int_x^{r_s} Y(r)\,\mathrm{d}r \right] \right\} = 2\exp\left[ 2\int_x^{r_s} Y(r)\,\mathrm{d}r \right] \frac{\mathrm{d}}{\mathrm{d}x}\left[ \int_x^{r_s} Y(r)\,\mathrm{d}r \right]$$

$$= 2\exp\left[ 2\int_x^{r_s} Y(r)\,\mathrm{d}r \right] \frac{\mathrm{d}}{\mathrm{d}x}\left[ F(r_s) - F(x) \right]$$

$$= -2 Y(x) \exp\left[ 2\int_x^{r_s} Y(r)\,\mathrm{d}r \right]$$

where $F$ is the primitive of $Y$, it ensues:

$$\int_r^{r_s} Y(x)\, exp\left[ 2\int_x^{r_s} Y(r)\,\mathrm{d}r \right]\,\mathrm{d}x = -\frac{1}{2}\left\{ \exp\left[ 2\int_x^{r_s} Y(r)\,\mathrm{d}r \right] \right\}_r^{r_s}$$

$$= \frac{1}{2}\left\{ \exp\left[ 2\int_r^{r_s} Y(r)\,\mathrm{d}r \right] - 1 \right\}$$

Therefore:

$$\int_r^{r_s} W(x)\,\mathrm{d}x = \frac{c\tau}{4 f_r F_{cor}} \frac{W(r_s)}{LR_a(r_s)}\left\{ \exp\left[ 2\int_r^{r_s} Y(r)\,\mathrm{d}r \right] - 1 \right\}$$

Finally, the exponential term becomes:

$$\exp\left[ 2\int_r^{r_s} Y(r)\,\mathrm{d}r \right] = 1 + \frac{4 f_r F_{cor}}{c\tau} \frac{LR_a(r_s)}{W(r_s)}\left[ \int_r^{r_s} W(r)\,\mathrm{d}r \right]$$

*Competing interests.* The authors declare that they have no conflict of interest.

*Acknowledgements.* This research work has been performed within the framework of a CIFRE grant (ANRT) for the doctoral work of Florian Gaudfrin. The lidar systems has been funded by the PROMETE project (ONERA). The laser source used in this paper for the experimental setup has been designed by Benoit Faure, Paul-Henri Pioger, and Guillaume Huss from the company HORUS-LEUKOS.

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
