# Peer review of "A new lidar inversion method using a surface reference target. Application to the backcattering coefficient and lidar ratio retrievals of a fog-oil plume at short-range"

_Atmospheric Measurement Techniques, 2019_

## Referee Comment (RC1) · Anonymous Referee #1 · 13 Nov 2019

In this paper, the authors include the surface reflectance in the lidar equation and derive its inversion. The derivation of the inversion itself is new as far as I can tell. Section 2 includes the standard volumetric lidar equation with an additive term which was already derived in (Kavaya et al. AO 1983) or similar derivations (Josset et al. OE 2010). The inclusion of an additive term is trivial. The meaning of the equation and what is being done with it (determination of instrumental constant, lidar ratio retrieval) is not new either. This has been proposed and done by (O' Connor JAOT 2004, Hu IEEE GRSL 2007) with a different kind of target (water clouds). With a surface reference

target there's relevant discussions by (Josset et al. IEEE GRSL 2010, IEEE TGRS 2018). However, I believe there is value in the formal derivation of the inversion and that the methodology could be applied to a standardized calibration of lidar systems with more descriptions of the field experiment. I suggest a major review. The changes are suggest are not necessarily difficult to implement but I would like the authors to think carefully and take the time needed to present a significantly revised version of the manuscript.

Major comments: - Several key references are missing. This paper seems surprisingly out of context of relevant research. The inclusion of these references could strengthen its content. - - The description of the experiment (section 5) lacks details. It makes it harder to understand the domain of validity and application of the methodology. - - I believe that several statements made in the paper are wrong (see some minor comments). These statements are mostly in the context and perspectives so they do not directly impact the core of the presented work. It could be related to the lack of references.

Minor comments: p.2 line 27 I would suggest to add more references on lidar calibration based on molecular backscattering. One recent example related to the CALIPSO lidar would be (Kar et al. AMT 2018).

p.2 line 52 "It is worth indicating that coupling lidar and sunphotometer measurements is possible only daytime while Raman measurements are carried out preferentially at nighttime in order to increase the SNR."

This is correct but the statement does not apply to the work of (Hu et al. IEEE GRSL 2007, etc). Please rephrase after more references are added to the manuscript.

p. 3 line 76 I'm not sure that I understand why the SRT is assumed to be Lambertian here. As far as I can tell, the formalism you derive is valid for any surface reflectance. It is a wrong assumption to make that natural surfaces are Lambertian, please include (Breon et al. JGR 2002) in the references. Limiting the formalism to Lambertian sur-

faces seriously limits the usefulness of this research. If one of the equation explicitly requires the surface to be Lambertian. Please state it explicitly in the manuscript.

Eq. 1 I don't understand why Fcor is not applied to the volumetric target. Please clarify.

Eq. 1 Fcor definition is on page 4, please define quantities the first time they are used.

p.4 line 105 the statement is confusing (beta missing, definition of the lidar ratios lines 91 and 92 ok).

p.5 In Eq (8) it could help to clarify that Y(rs) = 0 (only surface at rs).

Eq (9) clarify that it applies only before the surface.

About my two previous comments: in general, it is not very clear that there are two separate domains (as a function of range) for the equation.

p. 6 I'm not sure if there's a typo in Eq (13) or if I'm missing something. Please rephrase the comment right before Eq (13). It could help to clarify the matter.

p. 16 line 242 How do you know the reflectance of the Lambertian surface ? More detail are needed to describe this experiment (see major comment). Please expand this section.

line 243 "than 100 signals in 0.1s (1 kHz)"

This is redundant. Please remove or rephrase.

p. 19 line 287 "Indeed, BRDF are often considered as Lambertian for natural targets (surface roughness, vegetation...), so it can be replaced by SRT reflectance."

I believe it is a wrong assumption, I'm again referring to (Breon et al. JGR 2002). Recent research could imply that using reference measurements out of the hotspot would be ok for a lidar (Josset et al. IEEE TGRS 2018) but please rephrase this statement.

---

## Referee Comment (RC2) · Anonymous Referee #2 · 3 Jan 2020

Review of "A new lidar inversion method using a surface reference target. Application to the backscattering coefficient and lidar ratio retrievals of a fog-oil plume at short range" by Gaudfrin Âňet al.

This manuscript presents a lidar inversion technique that uses a surface reference target (SRT) with known reflectance properties to directly derive particulate backscatter coefficient and lidar ratios without any assumptions about the optical properties of the plume. The manuscript is clear and well written, however, there a few specific comments that if addressed, would improve the manuscript. Therefore, I recommend

publication following minor revision.

Major Comments: 1. The author go through great detail deriving the new inversion technique and testing the sensitivity of the retrieval to both theoretical and real scenarios, however, the manuscript would greatly benefit from a more detailed discussion on the applications and limitations of this approach. For example, theoretical testing was performed for range r = 100 m. Is this approach applicable at r=200 m? What is your definition of "short range"? 2. The authors mention that this approach has the potential to be applied to airborne lidar observations, however, I do not see how this would be possible without a) flying low in the atmosphere and b) knowing the varying underlying surface BRDF.

Minor Comments: Line 11 – Consider omitting the 3 dots following "ocean" Line 54-55 – This limitation is only applicable for ground-based lidar systems. Equation 1 – Please provide a definition of Fcor in the equation description and also consider adding some text explaining the BRDF component (f). Line 123 – Typo - "mentioning" Line 139-140 – "A prioris...". Consider adding justification/references for this sentence.

AMTD

---

## Author Comment (AC2) · 20 Jan 2020

**Response to Referee #2**

Many thanks to referee for take in time to evaluate and improve this manuscript. Thank also for your recommendation of publication. Please find below point by point responsive to comments.

1) The author go through great detail deriving the new inversion technique and testing the sensitivity of the retrieval to both theoretical and real scenarios, however, the manuscript would greatly benefit from a more detailed discussion on the applications and limitations of this approach. For example, theoretical testing was performed for range r = 100 m. Is this approach applicable at r=200 m? What is your definition of "short range"?

Short range means herein measurement which are made at distance less than 1 km. The inversion formalism and the method for finding the lidar ratio described in this manuscript are applicable as soon as the surface echo is present on the lidar signal. Applications at 200 m 500 m or 1 km are therefore possible.

Sentence added (p14. Line243-246): "Note that the formalism and methodology adopted here to retrieve the lidar ratio are efficient as long as the peak backscattering of the SRT is present on the lidar signal. The method has been evaluated, in this manuscript, for short range around 100 m because our research focus on application at this range. However, the algorithm developed does not present any limit with respect to the range provided that measurements are made below 1 km of range (this value depends of the power of laser sources) with respect to our applications. However, at first sight there is no limit to the application of the method to measurements at longer ranges such as more than 1 km measurements."

2) The authors mention that this approach has the potential to be applied to airborne lidar observations, however, I do not see how this would be possible without a) flying low in the atmosphere and b) knowing the varying underlying surface BRDF.

When we talk about airborne lidar observation, we are indeed considering low-level flights operated from aircraft such as helicopters or small planes. Precisely, measurements must be made at 1 km of altitude maximum. We suggest coupling the measurements with those of a spectrometer imager in order to deduce the reflectance of the surface target.

Sentence revised (P19. line 322-324): "The new inversion technique presented in this paper suggests new airborne lidar applications operated at low altitude from aircraft (helicopters, airplanes), but requires a priori knowledge of the reflectance of the SRT."

Paraph revised (p19. line 324-328): "Even if some models exist for the BRDF of surfaces (Bréon et al., 2002; Lobell and Asner, 2002; Mishchenko et al., 1999), their use seems difficult to implement because of the diversity of encountered surfaces during airborne measurements. Nevertheless, it may be possible to identify the reflectance of the ground surface by means of a spectroradiometer imager (Josset et al., 2018; Miesch et al., 2005; Poutier et al., 2002). The combination of these results measurements with the herein proposed inversion method would be a priori complementary to establish new methods of calibration for downlooking lidar measurements (spaceborne or airborne lidars)."

**3) Line 11 – Consider omitting the 3 dots following "ocean"**

The three dots have been deleted (p1. Line11).

4) Line 54-55– This limitation is only applicable for ground-based lidar systems.

Sentence revised (p3. line 64-65): "Another limitation of ground-based lidar measurements is related to the overlap function that strongly impacts (and prevents) observation close to the instrument, i.e. in the lowest layers of the troposphere where aerosols are emitted."

5) Equation 1 – Please provide a definition of Fcor in the equation description and also consider adding some text explaining the BRDF component (f).

Fcor is now defined (p4. Line 99).

Sentence revised (p.3 line 85-86): "In our approach, we propose to use a SRT of known bidirectional reflectance distribution function (BRDF)  $f_{r,\lambda}$  (in  $sr^{-1}$ ) (Kavaya et al., 1983; Nicodemus, 1965)."

Sentence added (p4. Line94-96): "It should be noted that in the particular case of a Lambertian surface  $f_{r,\lambda}(r_s, \theta_i)$  can be easily expressed by spectral bidirectional reflectance factor  $\rho_{\lambda}$  from  $\rho_{\lambda} \cos \theta_i / \pi$  (Haner et al., 1998; Josset et al., 2010, 2018). However, the general form of BRDF  $(f_{r,\lambda})$  will be considered later in this work in order to not restrict the approach to specific cases."

**6) Line 123 – Typo - "mentioning"**

Sentence revised (p.6 line 140): "It is worth mentioning that  $LR_a(r_s)$  is the lidar ratio just before the SRT and  $Y(r_s) = 0$  (only surface target)."

7) Line 139-140 – "A prioris: : :". Consider adding justification/references for this sentence.

We understand that this sentence is a bit confusing. Also, it is not necessary for the scientific content. We have therefore deleted it.

---

## Author Comment (AC3) · 20 Jan 2020

**Response to Referee #1**

Many thanks to referee for take in time to evaluate and improve this manuscript. Thank also for your recommendation of publication. Please find below point by point responsive to comments.

1) In this paper, the authors include the surface reflectance in the lidar equation and derive its inversion. The derivation of the inversion itself is new as far as I can tell. Section 2 includes the standard volumetric lidar equation with an additive term which was already derived in (Kavaya et al. AO 1983) or similar derivations (Josset et al. OE 2010). The inclusion of an additive term is trivial. The meaning of the equation and what is being done with it (determination of instrumental constant, lidar ratio retrieval) is not new either. This has been proposed and done by (O' Connor JAOT 2004, Hu IEEE GRSL 2007) with a different kind of target (water clouds). With a surface reference target there's relevant discussions by (Josset et al. IEEE GRSL 2010, IEEE TGRS 2018). However, I believe there is value in the formal derivation of the inversion and that the methodology could be applied to a standardized calibration of lidar systems with more descriptions of the field experiment. I suggest a major review. The changes are suggest are not necessarily difficult to implement but I would like the authors to think carefully and take the time needed to present a significantly revised version of the manuscript. Several key references are missing. This paper seems surprisingly out of context of relevant research. The inclusion of these references could strengthen its content.

*The originality of our works lies in two points. Firstly, the use of a surface target as a reference in the formalism of lidar inversion commonly called Klett inversion. Second, a simple method to identify the lidar ratio with two lidar measurements from the echo on a surface target in the presence and absence of a volume target under study (without coupling to other instruments). Suggested comments and references (in the general and specific remarks) have been taken into account to better situate our work in relation to the state of the art.*

2) The description of the experiment (section 5) lacks details. It makes it harder to understand the domain of validity and application of the methodology.

*Paraph add (p16. Line 264-274):* *"In order to increase the SNR, we preprocess the measurements from three lidar measurements:*

> *Signal 1. The first measurement is made by occulting the emitted laser beam to get a measure of the background scene (contribution of passive illumination);*

> *Signal 2. The second measurement is made by occulting the telescope to estimate the electronic noise of darkness of the instrument;*

> *Signal 3. The last measurement is made without any occultation.*

*For a given acquisition period, these three series of measured signals are averaged. The average signals of the background radiation and the electronic noise (signals 1 and 2) are subtracted from the signal 3."*

*Paragraph revised (p.17 line 278-282):* *"The high-speed sampling gives a measurement point every 5 cm on the line of sight of the lidar. Combined with a short pulse duration of the laser source (1.7 ns), this makes it possible to highlight local variations in concentration in the plume volume with the presence of two maxima at 38m and 39m. The backscatter peak of the surface target is also well sampled. The signal amplitude corresponding to the backscatter of the SRT is lower on $V_{sv}$ than on $V_s$ because of the presence of the oil plume."*

3) I believe that several statements made in the paper are wrong (see some minor comments). These statements are mostly in the context and perspectives so they do not directly impact the core of the presented work. It could be related to the lack of references.

*All the remarks and suggested references have been taken into account so that there is no longer any confusion.*

4) p.2 line 27 I would suggest to add more references on lidar calibration based on molecular backscattering. One recent example related to the CALIPSO lidar would be (Kar et al. AMT 2018).

*Sentence added (p2. line 28-30): "This calibration layer can be very high in altitude; it has recently been moved from the upper troposphere (30-34km) to the lower stratosphere (36-39km) for the CALIPSO spatial lidar to reduce inversion uncertainties (Getzewich et al., 2018; Kar et al., 2018)."*

5) p.2 line 52 "It is worth indicating that coupling lidar and sunphotometer measurements is possible only daytime while Raman measurements are carried out preferentially at nighttime in order to increase the SNR." This is correct but the statement does not apply to the work of (Hu et al. IEEE GRSL 2007, etc). Please rephrase after more references are added to the manuscript.

*Paragraph added (p3 line 56-63): "A fourth method consists in the determination of the optical thickness and lidar ratio of transparent layers located above opaque clouds (Hu et al., 2007; Young, 1995) that are used as reference for calibration in the inversion process (O'Connor et al., 2004). This method is used for downlooking lidar measurements capable of measuring depolarization ratios. However, the method is limited to lidar systems in non-polarized detection, or for lidar measurements for which clouds cannot be used as a reference. A fifth approach consists in the determination of the optical thickness of the atmosphere from the sea surface echo by combining lidar and radar measurements (Josset et al., 2008, 2010b). This method has been used to find the lidar ratio and the optical depth of aerosol layers over oceans (Dawson et al., 2015; Josset et al., 2012; Painemal et al., 2019)"*

*Sentence revised (p3. Line 79): "Also a new technique to retrieve the lidar ratio without using any sunphotometer, Raman or radar measurements is presented and applied to an aerosol plume."*

6) p. 3 line 76 I'm not sure that I understand why the SRT is assumed to be Lambertian here. As far as I can tell, the formalism you derive is valid for any surface reflectance. It is a wrong assumption to make that natural surfaces are Lambertian, please include (Breon et al. JGR 2002) in the references. Limiting the formalism to Lambertian surfaces seriously limits the usefulness of this research. If one of the equation explicitly requires the surface to be Lambertian. Please state it explicitly in the manuscript.

*Sentence revised (p.3 line 85-86): "In our approach, we propose to use a SRT of known bidirectional reflectance distribution function (BRDF) $f_{r,\lambda}$ (in $sr^{-1}$) (Kavaya et al., 1983; Nicodemus, 1965)."*

*Sentence added (p4. Line94-96): "It should be noted that in the particular case of a Lambertian surface $f_{r,\lambda}(r_s, \theta_i)$ can be easily expressed by spectral bidirectional reflectance factor $\rho_\lambda$ from $\rho_\lambda \cos \theta_i / \pi$ (Haner et al., 1998; Josset et al., 2010a, 2018). However, the general form of BRDF ($f_{r,\lambda}$) will be considered later in this work in order to not restrict the approach to specific cases."*

7) Eq. 1 I don't understand why Fcor is not applied to the volumetric target. Please clarify.

*Sentence add (p4. Line 102): "The factor does not apply to the volume part of the lidar equation, because in this expression the pulse profile is approximated to be constant over a rate duration $\tau_\lambda$. This approximation cannot be made on the backscatter peak of a surface target, because the backscattered energy is not integrated over a volume."*

8) Eq. 1 Fcor definition is on page 4, please define quantities the first time they are used.

We have moved Fcor definition the first time there is used (p4. Line 99)

9) p.4 line 105 the statement is confusing (beta missing, definition of the lidar ratios lines 91 and 92 ok).

We want to remove the extinction dependence of Eq 3. in order to solve it. For this we use the expression of the lidar ratio defined (now p4. line 112-113).

10) p.5 In Eq (8) it could help to clarify that Y(rs) = 0 (only surface at rs).

Sentence revised (p6. Line 140): *"It is worth mentioning that $LR_a(r_s)$ is the lidar ratio just before the SRT and $Y(r_s) = 0$ (only surface target)."*

11) Eq (9) clarify that it applies only before the surface. About my two previous comments: in general, it is not very clear that there are two separate domains (as a function of range) for the equation.

Sentence revised (p6. Line 143): *"This equation applies only before the surface target and can be solved by integrating both sides from $r$ to $r_s$ (Vande Hey, 2014)."*

12) p. 6 I'm not sure if there's a typo in Eq (13) or if I'm missing something. Please rephrase the comment right before Eq (13). It could help to clarify the matter.

Sentence revised (p6. Line 150-151): *"Multiplying the numerator and the denominator of the first term on the right-hand side of the subtraction by $D(r_s; 0)$, this expression becomes:"*

13) p. 16 line 242 How do you know the reflectance of the Lambertian surface? More detail are needed to describe this experiment (see major comment). Please expand this section.

Sentence added (p16. Line 263-265): *"A Lambertian Zenithal SRT (SphereOptics) with a $f_r = 0.20/\pi$ is placed at 52 m far away from the source. Its spectral bidirectional reflectance has been checked using laboratory bench measurements (Ceolato et al., 2012). The mean direction of the laser beam is parallel to the normal of the surface."*

14) line 243 "than 100 signals in 0.1s (1 kHz)" This is redundant. Please remove or rephrase.

Sentence revised (p16. Line 266-267): *"The laser source has repetition frequency of 1kHz. In order to increase the SNR, we preprocess the measurements from three lidar measurements:"*

15) p. 19 line 287 "Indeed, BRDF are often considered as Lambertian for natural targets (surface roughness, vegetation...), so it can be replaced by SRT reflectance." I believe it is a wrong assumption, I'm again referring to (Breon et al. JGR 2002). Recent research could imply that using reference measurements out of the hotspot would be ok for a lidar (Josset et al. IEEE TGRS 2018) but please rephrase this statement.

Paraph revised (p19. line 324-328): *"Even if some models exist to model the BRDF of surfaces (Bréon et al., 2002; Lobell and Asner, 2002; Mishchenko et al., 1999), their use seems difficult to implement, given the diversity of possible surfaces during airborne campaigns. Nevertheless, it may be possible to identify the reflectance of the ground surface from and the spectroradiometer imager (Josset et al., 2018; Miesch et al., 2005; Poutier et al., 2002). The combination of these results with the herein proposed inversion method would a priori be complementary to establish new methods of calibration for downlooking lidar measurements (spaceborne or airborne lidars)."*

---

## Author Response (AR2)

**Response to Editor**

Many thanks to Editor for take in time to evaluate and improve this manuscript. Please find below point by point responsive to comments.

1) p.4 L98. Please provide a reference related to the "conventionally a square-shaped pulse" as I am not familiar with this formalism. If the reference of interest is Paschotta 2008 you can disregard this comment but I expect that more than one reference would support a well established convention.

*The sentence has been modified and we have added a reference (p.4. line 98-99): $P_{p,\lambda}$ is a rectangular-shaped pulse in volumic lidar equation (Measures, 1992) viz. the ratio between the pulse energy and $\tau_\lambda$.*

2) p.6 L150. I think that just looking at the equations, I can understand how to go from Eq (12) to Eq (13) but the revised comment confuses me even more than the previous statement. The right hand side of the subtraction means βb(r) ? I think that it would be helpful to clarify the sentence or add more explanations.

*The sentence has been modified (p.6 line 150): "Multiplying the numerator and the denominator term of the fraction by $D(r_s, 0)$, this expression becomes:"*

3) p. 14 Fig. 6 Typo in "Theoritical"

*It is corrected. We have replaced "Theoritical" by "Theoretical" in legend graph Fig.6 and Fig. 7.*

4) p. 16 L270. I'm not sure I follow exactly what is done here so I elaborate on my understanding. When you occult the laser beam but the whole lidar is still on, then you measure both the background from the sun/artificial lights and the electronic noise. Consider that you could be in a dark room and switch bright lights on and off. When the light is off, you would measure the electronic noise (identical in principle to signal 2). When you switch the lights on, the electronic noise would not disappear. The way to combine electronic and background illumination could probably be the topic of a full technical report but considering an addition is acceptable. However, if you remove signal 1 and signal 2 from signal 3, you remove the electronic noise twice.

Please elaborate on this section and if necessary correct the data analysis procedure and the results.

*We estimate the electronic noise from signal 2. The measurement of signal 1 allows us to estimate the passive contribution from signal 1 and signal 2 by (signal 1 - signal 2).*

*In lidar measurements, we subtract the dark noise and the passive contribution from signals 1 and 2 by: signal 3 - (signal 1- signal 2). Thus, the dark noise is not subtracted twice. We have modified our paragraph to make this clearer.*

*The paraph has been revised p16. Line (274-276): "The averaged signals of the background radiation and the dark noise (signals 1 and 2) are then subtracted from the signal 3 such as: signal 3 - (signal 1 - signal 2)"*

5) p. 16 L272. You may consider to change "electronic noise of darkness" into "electronic noise" or "dark noise".

*The sentence has been revised (p.16 line 272): We have replaced "electronic noise" into "dark noise".*

6) p. 17 L280 – L281. Everything you say is true but this sentence could lead the reader to believe that the resolution of the system is 5 cm. The system can definitely detect the position of the plume within 5 cm, however the pulse duration will create correlations between the measurements in the order of 25 cm inside the plume. It's not a key point of your study but I would suggest to modify this sentence.

The sentence has been modified (p17. Line 280-282): *"Combined with a short pulse duration of the laser source (1.7 ns), this makes it possible to highlight local variations concentration in the order of 25 cm inside the plume with the presence of two maxima at 38 m and 39 m from lidar."*

7) p. 18 Table 4 L1. Unit should be sr not sr-1."

Exactly, it is corrected.